Association between risk factors and bone mineral density and the development of a self-assessment tool for early osteoporosis screening in postmenopausal women with type 2 diabetes

Chen Xiaoyu 1
Jia Xiufen 1
Lan Junping 1
Wu Wenjun 2
Ni Xianwu 1
Wei Yuguo 3
Zheng Xiangwu 1 zxwu111@sina.com
Liu Jinjin 1 jinjinliu@wmu.edu.cn
1 Department of Radiology, The First Affiliated Hospital of Wenzhou Medical University , Wenzhou , China
2 Department of Endocrinology, The First Affiliated Hospital of Wenzhou Medical University , Wenzhou , China
3 Advanced Application, Global Medical Service, GE Healthcare , Hangzhou , China
Menini Stefano
Electronic publication date: 2024 Oct 11
Publication date: 2024
Volume: 12
Electronic Location ID: e18283
Received 2024 Jun 10; Accepted 2024 Sep 19
Copyright: © 2024 Chen et al.
Copyright year: 2024
Copyright holder: Chen et al.
License: This is an open access article distributed under the terms of the Creative Commons Attribution License, which permits unrestricted use, distribution, reproduction and adaptation in any medium and for any purpose provided that it is properly attributed. For attribution, the original author(s), title, publication source (PeerJ) and either DOI or URL of the article must be cited.
License URL: https://creativecommons.org/licenses/by/4.0/

Keywords: Osteoporosis, Bone mineral density, Diabetes, Postmenopausal women, BMI, Machine learning, Dual-energy X-ray absorptiometry, Bone, Screening tool, Aging

Funding: Basic Research Project of Wenzhou Y2023191 Wenzhou Medical University 89622010 This work was supported by the Basic Research Project of Wenzhou (No. Y2023191) and the College-level Research Project of Wenzhou Medical University (No. 89622010). The funders had no role in study design, data collection and analysis, decision to publish, or preparation of the manuscript.

==============================
Background

Both diabetes and osteoporosis have developed into major global public health problems due to the increasing aging population. It is crucial to screen populations at higher risk of developing osteoporosis for disease prevention and management in postmenopausal women with type 2 diabetes (T2D). This study aims to quantitatively investigate the association between risk factors and bone mineral density (BMD) and develop a self-assessment tool for early osteoporosis screening in postmenopausal women with T2D.

Methods

We retrospectively enrolled 1,309 postmenopausal women with T2D. Linear regression methods were used to assess the association between risk factors and BMD. Additionally, a multivariate logistic regression analysis was performed to identify independent risk factors associated with osteoporosis. Utilizing the logistic regression machine learning algorithm, we developed an osteoporosis screening tool that categorizes the population into three risk regions based on age and body mass index (BMI), indicating low, moderate, and high prevalence of osteoporosis in the age-BMI plane.

Results

Older age and lower BMI were independently associated with decreased BMD. The BMD at the total hip, femur neck, and lumbar spine differed by 12.9, 10.9, and 15.5 mg/cm2 for each 1 unit increase in BMI, respectively. Both age and BMI were identified as independent predictors of osteoporosis. The osteoporosis screening tool was developed by using two straight lines with equations of BMI = 0.56 * age−4.12 and BMI = 0.56 * age−10.88; there were no significant differences in the prevalence of osteoporosis among the training, internal test, and external test datasets in the low-, moderate-, and high-risk regions.

Conclusion

We have successfully developed and validated a self-assessment tool for early osteoporosis screening in postmenopausal women with T2D for the first time. BMI was identified as a significant modifiable risk factor. Our study may improve awareness of osteoporosis and is valuable for disease prevention and management for postmenopausal women with T2D.

Introduction

Both diabetes and osteoporosis have emerged as significant global public health problems, with a growing prevalence attributed to the aging population (Camacho et al., 2020; Schacter & Leslie, 2021; Sun et al., 2022; Wang et al., 2021). According to the International Diabetes Federation (Sun et al., 2022), the global diabetes prevalence in adults aged between 20 and 79 years was approximately 10.5% in 2021 and is projected to increase to 12.2% by 2045. Type 2 diabetes (T2D) is the primary type of diabetes, accounting for over 90% of all diabetic cases. The percentage of adults with diabetes rises with age, reaching 29.2% among those ≥65 years old, according to the United States National Diabetes Statistics Report 2020 (Centers for Disease Control and Prevention, 2022). Osteoporosis, diagnosed through bone mineral density (BMD) measurement, is one of the most prevalent metabolic skeletal disorders characterized by reduced bone mass and increased destruction of bone microstructure, ultimately leading to low-impact fragility fracture (Afarideh, Sartori-Valinotti & Tollefson, 2021; Curry et al., 2018). The economic burden of osteoporosis-related fractures amounted to approximately $17.9 billion in the USA and £4 billion in the UK annually (Clynes et al., 2020). A meta-analysis (Liu et al., 2023) revealed a prevalence of osteoporosis at 27.67% among individuals with T2D. Diabetes is linked to lower BMD and an elevated risk of osteoporosis, especially in postmenopausal women with T2D, yet there remains limited understanding regarding the effects of risk factors on BMD. It is crucial to screen populations at increased risk for developing osteoporosis and fracture among postmenopausal women with T2D for effective disease prevention and management in clinical practice (Cairoli et al., 2023). Unfortunately, a survey among US postmenopausal women found that 44% had not undergone screening for osteoporosis, while 24% believed it was impossible to build new bone at their age, indicating a lack of awareness about the diagnosis of osteoporosis and osteoporotic fracture risk among this demographic (Lewiecki et al., 2019).

Several assessment tools have been developed for osteoporosis screening, including the Fracture Assessment Tool (FRAX) (Kanis; Kanis et al., 2008), Osteoporosis Self-assessment Tool for Asians (OSTA) (Koh et al., 2001), Simple Calculated Osteoporosis Risk Estimation Tool (SCORE) (Cadarette, Jaglal & Murray, 1999; Cadarette et al., 2001), osteoporosis tool for Chinese women (OSTC) (Tang et al., 2020), and the Study of Osteoporotic Fractures (SOF)-based Screening Tool (Brenneman et al., 2003). However, there is currently no optimal osteoporosis risk assessment tool for identifying low BMD (Crandall, 2015) designed specifically for postmenopausal women with T2D, and quantitative relationships between risk factors and BMD in this population have not been well studied.

Therefore, the objective of this study was to quantitatively investigate the correlation between risk factors and BMD, as well as to develop a self-assessment tool for early osteoporosis screening in postmenopausal women with T2D. Our findings may contribute to raising awareness about osteoporosis and assisting in its prevention and management among postmenopausal women with T2D.

Materials and Methods

Patients

Ethical approval was obtained from the ethics committee of the First Affiliated Hospital of Wenzhou Medical University (No. 2022-r064) for this study, and informed consent requirement was waived due to the retrospective design. Postmenopausal women with T2D were enrolled at the the First Affiliated Hospital of Wenzhou Medical University from January 2020 to June 2022 and at the Second Affiliated Hospital of Wenzhou Medical University from October 2022 to December 2022. All patients underwent dual-energy X-ray absorptiometry (DXA) scans. Exclusion criteria included history or evidence of hyper- or hypoparathyroidism, cancers, hysterectomy or postoperatively menopause, hip or joint replacement, use of estrogens, steroids, corticosteroids, bisphosphonates, and calcitonin drugs, and missing most medical records. Body mass index (BMI) was calculated as weight/height2.

BMD measurements

DXA (Luna Prodigy Advance; GE Medical Systems Ultrasound & Primary Care Diagnostics, LLC, Madison, WI, USA) was employed to measure BMD in the anteroposterior lumbar spine and left hip. In cases where the left hip had been previously fractured, measurements were taken from the right side. T-scores were calculated using the normal reference values from an age- and gender-matched Chinese group provided by the DXA system manufacturer, which was given by Kan et al. (2021)

(1) T=(X−μref)/SDref,

where X represents the observed BMD of a patient, μref is the BMD in young adults, and SDref refers to the standard deviation of BMD in young adults. The region with the lowest T-score from either the lumbar spine (L1–L4) or hip at the femoral neck or total hip was used to classify BMD. osteoporosis was defined as a T-score less than or equal to 2.5 at the lumbar spine, femoral neck, and total hip (Kanis et al., 1994).

Development of a screening tool

Consider two clinical features, x1 and x2 in a Cartesian coordinate system (see Fig. 1), the coordinates of each patients with features of x1 and x2 can be plotted in this coordinate system. A straight line in this coordinate system is given by

(2) x2=kx1+c,

where k and c are the slope and intercept of the line, respectively. Two distinct straight lines with varying values of c can partition the x1 and x2 points from all patients into three regions. We categorized patients into two groups: osteoporosis and non-osteoporosis; the former includes osteoporosis patients, while the latter encompasses normal and osteopenia patients. The two straight lines with identical slope k and different values of c can divide patients into three risk regions representing low, moderate, and high prevalence of osteoporosis, corresponding to low-, medium- and high-risk levels for developing osteoporosis.

Figure 1 Different risk regions divided by two straight lines in the x1–x2 Cartesian coordinate system.

In this study, the value of k was determined using a logistic regression machine learning algorithm. The Coding was implemented in Python 3.7 and Scikit-learn 1.0.2. The model parameters were set as follows: ‘C’, 1.0; ‘solver’, ‘lbfgs’; ‘penalty’, ‘l2’; and ‘max_iter’, 100. If we named the model as ‘LRmodel’ in Python, then k can be calculated as

(3) k=−LRmodel.coef[0,0]/LRmodel.coef[0,1],

where LRmodel.coef[0,0] and LRmodel.coef[0,1] represent the coefficients of the features x1 and x2 in the decision function after fitting the training datasets, respectively.

The risk of osteoporosis was determined by its prevalence. To the best of our knowledge, criteria for low, moderate, and high risk of osteoporosis have not been reported for postmenopausal women with T2D. Koh et al. (2001) proposed criteria for these categories without considering the factor of diabetes, finding prevalence rates of osteoporosis 1%, 10%, and 44% in the Japanese population and 3%, 15%, and 61% for the Asian people. In this study, we adopted the average prevalence of osteoporosis in the Japanese and Asian people at low and high risk, i.e., 2% for low risk and 52.5% for high risk. The prevalence in regions with low and high risk was adjusted as closely approximate 2% and 52.5% by modifying c values in the two straight lines given by Eq. (2). One can define other prevalence based on our dataset provided in the supplemental ‘excel’ file.

To validate the screening tool, we chose patients in the First Affiliated Hospital of Wenzhou Medical University between December 2021 and June 2022 and those in the Second Affiliated Hospital of Wenzhou Medical University from October 2022 to December 2022 as internal and external testing datasets, respectively.

Statistical analysis

Statistical analyses were performed using SPSS 22.0 (IBM SPSS, Armonk, NY, USA) and R 3.6.1 (R Core Team, 2019). Continuous variables were expressed as mean ± standard deviation and categorical variables as frequency (percentage). Univariate and multivariate linear regressions were utilized to assess the association between risk factors and BMD. Student t-test or Mann-Whitney U test were applied for comparing continuous variables, while χ2 tests or Fisher exact tests were used for comparing categorical variables. A forward stepwise multivariate logistic regression analysis was carried out to identify risk factors independently associated with osteoporosis. Factors with a P value less than 0.1 in the univariate analysis were included in the multivariate logistic regression analysis. A P value of less than 0.05 was considered statistically significant.

Results

One thousand three hundred and nine patients were included in this study, with 915 from the training dataset, 254 from the internal test dataset, and 140 from the external test dataset. The average age, height, weight, and BMI were 66.7 ± 8.4 years, 156.2 ± 5.7 cm, 58.3 ± 9.3 kg, and 23.9 ± 3.5, respectively (refer to Table 1). Osteoporosis was present in 546 (41.7%) of the patients.

Table 1 Baseline characteristics.

Variables	Total (n = 1309)	Training dataset (n = 915)	Internal test dataset (n = 254)	External test dataset (n = 140)	P	
Age (years)	66.7 ± 8.4	66.2 ± 8.2	68.8 ± 8.7	65.7 ± 8.1	<0.001	
Height (cm)	156.2 ± 5.7	156.6 ± 5.5	154.2 ± 6.1	156.7 ± 5.9	<0.001	
Weight (kg)	58.3 ± 9.3	58.2 ± 9.2	58.1 ± 8.7	59.4 ± 10.5	0.329	
BMI	23.9 ± 3.5	23.7 ± 3.5	24.4 ± 3.4	24.2 ± 3.9	0.006	
Number of osteoporosis patients	546 (41.7%)	372 (40.7)	103 (40.6%)	71 (50.7%)	0.073	

Table 2 illustrates the correlation between the risk factors and BMD in the training dataset. Older age, longer menopause years, and higher parity were associated with decreased BMD across all skeletal sites. Conversely, increased body weight, lower weight, lower BMI, reduced serum P and the presence of fatty liver were associated with lower BMD at all skeletal sites. Table 3 presents the multivariable model of the associations between risk factors and BMD in the training dataset. Age, BMI, fatty live disease, and atherosclerosis were independently correlated with total hip BMD. Total hip BMD differed by −7.2 mg/cm2 (95% CI [−8.2 to −6.1]; P < 0.001) for each one year increase in age, 12.9 mg/cm2 (95% CI [10.3–15.5]; P < 0.001) for each 1 unit higher BMI, 28.3 mg/cm2 (95% CI [9.8–46.7]; P = 0.003) in the presence of fatty liver disease, and −21.9 mg/cm2 (95% CI [−40.1 to −3.8; P = 0.018) in the presence of atherosclerosis. BMDs at femur neck and lumbar spine differed by 10.9 mg/cm2 (95% CI [8.5–13.4]; P < 0.001) and 15.5 mg/cm2 (95% CI [12.7–18.3]; P < 0.001) for every 1 unit increase in BMI, and by −7.3 mg/cm2 (95% CI [−8.3 to −6.2]; P < 0.001) and −7.1 mg/cm2 (95% CI [−8.3 to −6.0]; P < 0.001) for each one year increase in age, respectively. Age, BMI, and atherosclerosis were independently associated with the femur neck BMD. Older age and lower BMI were independently associated with lower lumbar spine BMD; similar associations were found for L1 to L4 BMD. It was noted that the number of children was significantly associated with BMD at all skeletal in univariate analyses; however, it was not independently associated with BMD at all skeletal sites in multivariate analyses.

Table 2 Associations between risk factors and BMD at total hip, femoral neck and lumbar spine L1–L4 in univariate linear regression analysis.

	Total hip BMD (mg/cm2)	Femoral neck BMD (mg/cm2)	Lumbar spine BMD (mg/cm2)	L1 BMD (mg/cm2)	L2 BMD (mg/cm2)	L3 BMD (mg/cm2)	L4 BMD (mg/cm2)	
	Estimate (95% CI)	P value	Estimate (95% CI)	P value	Estimate (95% CI)	P value	Estimate
(95% CI)	P value	Estimate (95% CI)	P value	Estimate (95% CI)	P value	Estimate (95% CI)	P value	
Age (years)	−7.2 [−8.3 to −6.1]	<0.001	−7.4 [−8.4 to −6.3]	<0.001	−6.7 [−7.9 to −5.4]	<0.001	−5.8 [−7.0 to −4.6]	<0.001	−6.9 [−8.2 to −5.6]	<0.001	−7.3 [−8.7 to −5.9]	<0.001	−5.5 [−6.9 to −4.0]	<0.001	
Height (cm)	5.7 [1.0–7.5]	<0.001	6.4 [4.8–8.1]	<0.001	6.2 [4.3–8.1]	<0.001	5.4 [3.6–7.3]	<0.001	6.6 [4.6–8.6]	<0.001	7.2 [5.1–9.4]	<0.001	6.3 [4.1–8.4]	<0.001	
Weight (kg)	5.9 [4.9–6.9]	<0.001	5.0 [4.0–5.9]	<0.001	6.4 [5.3–7.5]	<0.001	5.3 [4.3–6.4]	<0.001	6.6 [5.4–7.7]	<0.001	6.8 [5.6–8.1]	<0.001	6.7 [5.5–8.0]	<0.001	
BMI	12.4 [9.7–15.1]	<0.001	9.3 [6.6–11.9]	<0.001	13.9 [10.9–16.8]	<0.001	11.2 [8.3–14.1]	<0.001	13.9 [10.8–17.1]	<0.001	14.1 [10.7–17.5]	<0.001	14.6 [11.2–17.9]	<0.001	
Duration of diabetes	−1.5 [−2.8 to −0.3]	0.017	−1.9 [−3.1 to −0.7]	0.002	0.3 [−1.1 to 1.7]	0.675	0.1 [−1.2 to 1.5]	0.837	0.4 [−1.0 to 1.9]	0.575	1.2 [−0.3 to 2.8]	0.107	0.9 [−0.7 to 2.5]	0.266	
Hypertension	−5.3 [−25.9 to 15.3]	0.614	−21.0 [−41.0 to −1.1]	0.039	6.6 [−16.0 to 29.1]	0.568	7.7 [−14.0 to 29.4]	0.489	5.6 [−18.1 to 29.2]	0.644	8.6 [−16.9 to 34.1]	0.509	13.0 [−12.4 to 38.5]	0.315	
Smoking	−59.9 [−179.0 to 59.2]	0.325	−91.7 [−205.3 to 21.8]	0.114	1.3 [−132.4 to 135.1]	0.984	8.9 [−118.8 to 136.7]	0.891	−50.8 [−191.4 to 89.9]	0.479	13.8 [−135.2 to 162.9]	0.856	40.4 [−110.2 to 191.1]	0.599	
Alcohol	−16.3 [−85.6 to 52.9]	0.644	−18.0 [−84.0 to 48.0]	0.593	13.0 [−64.7 to 90.7]	0.743	−22.8 [−96.9 to 51.4]	0.548	5.7 [−76.1 to 87.5]	0.891	37.1 [−49.5 to 123.6]	0.402	22.5 [−64.9 to 110.0]	0.614	
Age at menopause	−0.5 [−3.7 to 2.6]	0.740	0.2 [−2.8 to 3.3]	0.874	−2.5 [−6.0 to 0.9]	0.151	−0.7 [−4.0 to 2.7]	0.692	−1.8 [−5.5 to 1.8]	0.318	−1.9 [−5.8 to 2.0]	0.347	−2.5 [−6.4 to 1.4]	0.209	
Years after menopause	−6.6 [−7.7 to −5.5]	<0.001	−6.8 [−7.8 to −5.8]	<0.001	−5.8 [−7.0 to −4.6]	<0.001	−6.5 [−6.5 to −4.1]	<0.001	−6.1 [−7.3 to −4.8]	<0.001	−6.5 [−7.8 to −5.1]	<0.001	−4.8 [−6.2 to −3.4]	<0.001	
Reproductive history	43.1 [−127.7 to 213.9]	0.621	100.9 [−64.8 to 266.6]	0.233	73.2 [−115.3 to 261.6]	0.447	78.6 [−102.6 to 259.9]	0.395	46.9 [−150.8 to 244.7]	0.642	85.9 [−126.7 to 298.6]	0.429	95.6 [−116.7 to 307.9]	0.378	
Number of children	−25.2 [−32.6 to −17.8]	<0.001	−23.1 [−32.3 to −17.9]	<0.001	−26.9 [−35.1 to −18.7]	<0.001	−22.1 [−29.9 to −14.3]	<0.001	−27.6 [−36.3 to −19.0]	<0.001	−29.0 [−38.3 to −19.8]	<0.001	−20.2 [−29.5 to −10.8]	<0.001	
History of fracture	−50.0 [−102.8 to 2.8]	0.064	−44.6 [−94.9 to 5.7]	0.083	−47.2 [−106.9 to 12.5]	0.122	−38.4 [−95.1 to 18.3]	0.185	−60.2 [−123 to 2.6]	0.061	−41.7 [−108.4 to 25.0]	0.221	−50.1 [−117.8 to 17.5]	0.147	
Diabetic retinopathy	−21.6 [−41.2 to −2.0]	0.031	−26.6 [−45.6 to −7.5]	0.006	−1.3 [−23.0 to 20.4]	0.906	−2.1 [−23.0 to 18.7]	0.840	−5.8 [−28.5 to 17.0]	0.619	12.1 [−12.3 to 36.6]	0.331	7.5 [−17.0 to 31.9]	0.549	
Diabetic peripheral neuropathy	7.6 [−12.0 to 27.1]	0.449	6.9 [−12.1 to 25.9]	0.479	28.3 [6.8–49.8]	0.010	22.8 [2.1 to 43.5]	0.031	28.0 [5.5–50.6]	0.015	32.5 [8.3–56.8]	0.009	36.3 [12.1 to 60.5]	0.003	
atherosclerosis	−47.8 [−67.4 to −28.2]	<0.001	−49.3 [−68.4 to −30.3]	<0.001	−29.7 [−51.5 to −7.9]	0.008	−31.8 [−52.7 to −10.8]	0.003	−37.7 [−60.5 to −14.9]	0.001	−34.0 [−58.6 to −9.4]	0.007	−10.3 [−34.9 to 14.4]	0.414	
Kidney disease	−37.1 [−66.1 to −8.2]	0.012	−42.6 [−70.6 to −14.5]	0.003	−19.1 [−51.1 to 12.9]	0.242	−12.0 [−42.8 to 18.8]	0.445	−6.9 [−40.5 to 26.7]	0.687	−17.8 [−53.9 to 18.3]	0.334	29.9 [−65.9 to 6.2]	0.105	
Fatty liver	55.5 [35.5–75.4]	<0.001	33.5 [13.9 to 53.1]	0.001	43.2 [21.0–65.3]	<0.001	36.8 [15.4 to 58.2]	0.001	40.2 [16.9–3.5]	0.001	35.2 [10.1–60.3]	0.006	48.4 [23.4 to 73.4]	<0.001	
Hypertension	−2.6 [−22.5 to 17.3]	0.797	−16.8 [−36.0 to 2.5]	0.088	6.3 [−15.6 to 28.2]	0.574	9.1 [−11.9 to 30.1]	0.397	4.4 [−18.5 to 27.4]	0.705	8.8 [−15.9 to 33.5]	0.486	12.2 [−12.5 to 36.8]	0.333	
Thyroid nodule	2.7 [−16.9 to 22.3]	0.786	9.0 [−10.1 to 28.0]	0.357	9.7 [−11.9 to 31.4]	0.377	6.2 [−14.6 to 27.0]	0.557	6.7 [−16.0 to 29.4]	0.561	9.0 [−15.4 to 33.4]	0.471	9.3 [−15.0 to 33.7]	0.454	
Poorer glycaemic control	40.0 [11.5–68.5]	0.006	36.4 [8.7 to 64.2]	0.010	27.1 [−4.4 to 58.6]	0.092	32.5 [2.2–62.8]	0.036	27.5 [−5.6 to 60.6]	0.103	16.9 [−18.7 to 52.6]	0.352	33.5 [−2.0 to 69.0]	0.065	
Diabetic pedipathy	−62.2 [−106.3 to −18.1]	0.006	−49.1 [−92.0 to −6.2]	0.025	−38.8 [−87.5 to 10.0]	0.119	−25.5 [−72.4 to 21.4]	0.287	−18.3 [−69.5 to 32.9]	0.483	−48.5 [−103.5 to 6.5]	0.084	−51.0 [−105.8 to 3.9]	0.069	
Hypovitaminosis	36.0 [−29.3 to 101.2]	0.280	31.2 [−32.1 to 94.6]	0.334	14.0 [−58.0 to 85.9]	0.703	54.0 [−15.1 to 123.1]	0.126	10.6 [−64.8 to 86.1]	0.783	−8.0 [−89.2 to 73.2]	0.847	−5.7 [−75.3 to 86.7]	0.891	
Alkaline phosphatase (U/L)	−0.1 [−0.4 to 0.1]	0.216	−0.1 [−0.4 to 0.1]	0.171	0.0 [−0.3 to 0.2]	0.753	0.0 [−0.3 to 0.2]	0.840	0.0 [−0.3 to 0.2]	0.799	0.0 [−0.3 to 0.3]	0.998	0.0 [−0.3 to 0.2]	0.828	
Ca (mmol/L)	106.7 [26.5–186.8]	0.009	83.6 [−5.5 to 161.8]	0.036	84.3 [−4.0 to 172.6]	0.062	49.9 [−35.2 to 134.9]	0.251	71.3 [−21.9 to 164.5]	0.134	100.8 [0.9–200.6]	0.048	79.8 [−19.9 to 179.4]	0.117	
P (mmol/L)	100.6 [52.9–148.3]	<0.001	100.8 [54.4 to 147.2]	<0.001	121.5 [69.2–173.8]	<0.001	111.8 [61.4–162.3]	<0.001	131.3 [76.1–186.4]	<0.001	145.0 [85.9–204.1]	<0.001	96.2 [36.9–155.5]	0.002	
HbAlc (%)	−1.0 [−5.6 to 3.6]	0.667	1.6 [−2.9 to 6.2]	0.483	2.0 [−3.1 to 7.0]	0.448	0.9 [−3.9 to 5.8]	0.707	2.2 [−3.1 to 7.5]	0.420	1.8 [−4.0 to 7.5]	0.544	2.7 [−3.0 to 8.4]	0.358	
PINP (μg/L)	−0.3 [−0.6 to 0.0]	0.046	0.0 [−0.3 to 0.2]	0.720	−0.2 [−0.5 to 0.2]	0.323	0.1 [−0.3 to 0.4]	0.730	0.0 [−0.3 to 0.4]	0.994	0.0 [−0.3 to 0.4]	0.840	−0.2 [−0.6 to 0.2]	0.325	
PICP (μg/L)	−39.2 [−81.5 to 3.0]	0.070	−7.4 [−47.5 to 32.7]	0.718	−55.2 [−103.2 to −7.2]	0.025	−13.8 [−81.5 to 31.6]	0.551	−29.9 [−80.9 to 21.1]	0.252	−27.1 [−83.1 to 28.9]	0.344	−63.7 [−118.5 to 8.9]	0.023	
Vitamin D (mmol/L)	0.6 [0.0–1.2]	0.071	0.7 [0.2 to 1.3]	0.013	0.2 [−0.5 to 0.8]	0.558	0.1 [−0.6 to 0.7]	0.791	0.0 [−0.7 to 0.7]	0.071	0.3 [−0.4 to 1.0]	0.432	0.3 [−0.4 to 1.1]	0.378	
Thyroxin (nmol/L)	−0.5 [−0.9 to 0.0]	0.046	−0.5 [−1.0 to 0.1]	0.027	0.0 [−0.5 to 0.5]	0.993	0.2 [−0.3 to 0.7]	0.469	0.1 [−0.4 to 0.6]	0.752	0.0 [−0.6 to 0.5]	0.928	−0.3 [−0.8 to 0.3]	0.326	
Glucocorticoid (nmol/L)	−0.1 [−0.3 to 0.0]	0.038	−0.1 [−0.2 to 0.0]	0.122	−0.1 [−0.2 to 0.1]	0.405	0.0 [−0.1 to 0.1]	0.931	0.0 [−0.1 to 0.2]	0.707	−0.1 [−0.3 to 0.0]	0.180	−0.1 [−0.2 to 0.1]	0.264	

Table 3 Multivariable models of the association between risk factors and BMD.

	Estimate (95% confidence interval)	P value	
Total hip BMD (mg/cm2)			
Age/year	−7.2 [−8.2 to −6.1]	<0.001	
BMI/1 unit	12.9 [10.3–15.5]	<0.001	
Fatty liver disease	28.3 [9.8–46.7]	0.003	
Atherosclerosis	−21.9 [−40.1 to −3.8]	0.018	
Femur neck BMD (mg/cm2)			
Age/year	−7.3 [−8.3 to −6.2]	<0.001	
BMI/unit	10.9 [8.5–13.4]	<0.001	
Atherosclerosis	−18.2 [−36.2 to −0.2]	0.047	
Lumbar spine BMD (mg/cm2)			
Age/year	−7.1 [−8.3 to −6.0]	<0.001	
BMI/1 unit	15.5 [12.7–18.3]	<0.001	
L1 BMD (mg/cm2)			
Age/1 year	−6.3 [−7.4 to −5.1]	<0.001	
BMI/1 unit	12.6 [9.8–15.3]	<0.001	
L2 BMD (mg/cm2)			
Age/1 year	−7.4 [−8.6 to −6.1]	<0.001	
BMI/1 unit	15.6 [12.6–18.5]	<0.001	
L3 BMD (mg/cm2)			
Age/1 year	−7.8 [−9.1 to −6.4]	<0.001	
BMI/1 unit	15.9 [12.7–19.1]	<0.001	
L4 BMD (mg/cm2)			
Age/1 year	−5.9 [−7.3 to −4.6]	<0.001	
BMI/1 unit	15.9 [12.6–19.1]	<0.001	

Table 4 shows the characteristics of patients in the osteoporosis and non-osteoporosis groups in the training dataset. The duration of diabetes was 11.7 ± 7.8 years, and the years since menopause were 16.3 ± 8.6 years. Osteoporosis patients had higher age and years since menopause while having lower height, weight and BMI than non-osteoporosis patients (all P < 0.001). The BMDs of lumbar spine, femur neck, and total hip between osteoporosis and non-osteoporosis patients were 0.829 ± 0.094 vs. 1.069 ± 0.130, 0.692 ± 0.091 vs. 0.866 ± 0.135, and 0.741 ± 0.101 vs. 0.924 ± 0.133 g/cm2, respectively. More osteoporosis patients were in the presence of atherosclerosis and diabetic pedipathy. Serum P in non-osteoporosis patients was significantly higher than in osteoporosis patients. Osteoporosis patients had more children than non-osteoporosis patients (3.2 ± 1.3 vs. 2.7 ± 1.2; P < 0.001).

Table 4 Characteristics of patients in the osteoporosis and non-osteoporosis groups.

	Total
(n = 915)	Non-osteoporosis
(n = 543)	Osteoporosis
(n = 372)	P	
Age (years)	66.2 ± 8.2	64.0 ± 8.2	69.4 ± 7.1	<0.001	
Height (cm)	156.6 ± 5.5	157.5 ± 5.2	155.4 ± 5.7	<0.001	
Weight (kg)	58.2 ± 9.2	60.4 ± 8.8	55.0 ± 8.8	<0.001	
BMI	23.7 ± 3.5	24.3 ± 3.4	22.8 ± 3.6	<0.001	
Duration of diabetes	11.7 ± 7.8	11.6 ± 7.8	11.8 ± 7.8	0.633	
L1 BMD (g/cm2)	0.873 ± 0.160	0.954 ± 0.131	0.755 ± 0.120	<0.001	
L2 BMD (g/cm2)	0.937 ± 0.174	1.032 ± 0.139	0.798 ± 0.119	<0.001	
L3 BMD (g/cm2)	1.022 ± 0.188	1.124 ± 0.148	0.872 ± 0.131	<0.001	
L4 BMD (g/cm2)	1.041 ± 0.187	1.141 ± 0.150	0.895 ± 0.132	<0.001	
Lumbar spine BMD (g/cm2)	0.971 ± 0.166	1.069 ± 0.130	0.829 ± 0.094	<0.001	
Femur neck BMD (g/cm2)	0.795 ± 0.146	0.866 ± 0.135	0.692 ± 0.091	<0.001	
Total hip BMD (g/cm2)	0.849 ± 0.151	0.924 ± 0.133	0.741 ± 0.101	<0.001	
Hypertension	538 (58.8%)	324 (59.7%)	214 (57.5%)	0.855	
Smoking	6 (0.7%)	3 (0.6%)	3 (0.9%)	0.720	
Drinking	18 (2.2%)	12 (2.5%)	6 (1.7%)	0.417	
Age at menopause	49.9 ± 3.1	49.8 ± 3.4	50.0 ± 2.8	0.359	
Years since menopause	16.3 ± 8.6	14.1 ± 8.6	19.4 ± 7.5	<0.001	
Reproductive History	911 (99.7%)	543 (100%)	368 (99.2%)	0.131	
Number of children	2.9 ± 1.3	2.7 ± 1.2	3.2 ± 1.3	<0.001	
History of fracture	31 (4.4%)	16 (4.0%)	15 (5.0%)	0.514	
Diabetic retinopathy	405 (44.3%)	229 (42.2%)	176 (47.3%)	0.124	
Diabetic peripheral neuropathy	438 (47.9%)	271 (49.9%)	167 (44.9%)	0.136	
atherosclerosis	539 (58.9%)	296 (54.5%)	243 (65.3%)	0.001	
Kidney disease	119 (13.0%)	64 (11.8%)	55 (14.8%)	0.185	
Fatty liver	335 (36.6%)	222 (40.9%)	113 (30.4%)	0.001	
Thyroid nodule	418 (45.7%)	246 (45.3%)	172 (46.2%)	0.781	
Poorer glycaemic control	123 (13.4%)	76 (14.0%)	47 (12.6%)	0.553	
Diabetic pedipathy	47 (5.1%)	20 (3.7%)	27 (7.3%)	0.016	
Hypovitaminosis	21 (2.3%)	15 (2.8%)	6 (1.6%)	0.254	
Alkaline phosphatase (U/L)	89 ± 46	89 ± 50	89 ± 40	0.824	
Ca (mmol/L)	2.28 ± 0.12	2.28 ± 0.12	2.27 ± 0.13	0.071	
P (mmol/L)	1.22 ± 0.21	1.25 ± 0.20	1.19 ± 0.22	<0.001	
HbAlc (%)	9.4 ± 2.2	9.4 ± 2.2	9.4 ± 2.3	0.994	
PINP (μg/L)	51.87 ± 44.73	51.35 ± 43.59	52.53 ± 46.23	0.779	
PICP (μg/L)	0.437 ± 0.306	0.430 ± 0.333	0.445 ± 0.269	0.600	
Vitamin D (mmol/L)	56.81 ± 10.04	57.45 ± 18.38	55.91 ± 19.92	0.312	
Thyroxin (nmol/L)	115.52 ± 24.35	114.03 ± 24.76	117.70 ± 23.63	0.049	
Glucocorticoid (nmol/L)	291 ± 128	282 ± 120	302 ± 137	0.164	

The multivariable logistic regression analysis (Table 5) revealed two independent predictors of osteoporosis: age (OR, 1.109; 95% CI [1.087–1.131]; P < 0.001) and BMI (OR, 0.828; 95% CI [0.790–0.867]; P < 0.001).

Table 5 Results of multivariate logistic regression analysis.

Variable	Β coefficient*	OR	95% CI	P value	
Age	0.103 ± 0.010	1.109	[1.087–1.131]	<0.001	
BMI	−0.189 ± 0.024	0.828	[0.790–0.867]	<0.001	
Notes:

OR, odds ratio; CI, confidence interval; BMI, body mass index.

* Values are means ± standard errors.

The scatter plot of BMI vs. age for patients in the training dataset is presented in Fig. 2A. Osteoporosis patients gradually increased from the upper left corner to the lower right corner on the BMI-age plane. Similar trends were found for the internal and external test datasets (see Figs. 2B and 2C). The slope k was determined to be 0.56 from the logistic regression machine learning algorithm. Two straight lines divided the BMI-age plane into three regions, with the low-risk region in the upper left, the moderate-risk region in the middle, and the high-risk region in the lower right, ensuring that the prevalence of osteoporosis in the low- and high-risk areas were as close as possible to 2% and 52.5%, respectively. The intercepts (i.e., cin Eq. (2)), of these two straight lines were determined to be −4.12 and −10.88. Therefore, the following equations describe the two straight lines shown in Fig. 2:

(4) BMI=0.56∗age−4.12

and

(5) BMI=0.56∗age−10.88.

Figure 2 Scatter plot of BMI-age and presentation of screen tool in the (A) training, (B) internal test and (C) external test datasets, and (D) bar plot of prevalence of osteoporosis in the low-, moderate- and high-risk regions among different datasets.

Note that the red and green points in circle shape on the age-BMI plane represent osteoporosis and non-osteoporosis samples. The age-BMI point coordinate in Fig. 2A can be determined based on a patient’s BMI value and age. If this point falls below the orange line with equation of BMI = 0.56 * age−10.88 in Fig. 2A, indicating that the patient’s BMI is less than 0.56 * age−10.88, then the patient is in the high-risk area for osteoporosis; if this point lies above the blue line in the equation of BMI = 0.56 * age−4.12 in Fig. 2A, suggesting that the patient’s BMI is greater than 0.56 * age−4.12, then the patient is categorized as being in the low-risk area for osteoporosis; otherwise, the patient falls into the medium-risk zone for osteoporosis.

The detailed method on how to use these two equations to evaluate the osteoporosis is described in the legend of Fig. 2.

Figure 2D shows the prevalence of osteoporosis as determined by the developed screening tool. The prevalence rates were 2.6%, 11.1%, and 11.1% for patients at low risk, 16.2%, 21.0%, and 26.8% for patients at moderate risk, and 52.6%, 48.1%, and 58.1% for patients at high risk in the training, internal test and external test datasets, respectively. It is important to note that there was no significant difference in the prevalence of osteoporosis at different risk categories across the training, internal test, and external test datasets (P = 0.232 for low-risk region; P = 0.220 for moderate-risk region; P = 0.254 for high-risk region; see Table 6).

Table 6 Number of patients in the low-, moderate- and high-risk regions between osteoporosis and non-osteoporosis groups.

	Non-osteoporosis	Osteoporosis	P	
Low-risk region			0.232	
Training dataset	38	1		
Internal test dataset	8	1		
External test dataset	8	1		
Moderate-risk region			0.220	
Training dataset	207	40		
Internal test dataset	49	13		
External test dataset	30	11		
High-risk region			0.254	
Training dataset	298	331		
Internal test dataset	95	88		
External test dataset	44	61		

Discussion

In this study, we investigated the influence of risk factors on BMD and proposed a simple osteoporosis self-assessment tool for postmenopausal women with T2D. To our knowledge, this is the first osteoporosis self-assessment tool specifically designed for this population.

We found the incidence of osteoporosis in 1,309 postmenopausal women with T2D was 41.7%. Other studies reported corresponding incidents of 30.40% (Viégas et al., 2011), 39% (Karimifar et al., 2012), 51.55% (Jianbo et al., 2014), and 39.63% (Li et al., 2021) in samples of 148, 200, 258, and 164 patients respectively. These differences in prevalence may be attributed to variations in populations, sample size, and methods for evaluating osteoporosis. It is evident that postmenopausal women with T2D are at a heightened risk for osteoporosis compared to non-diabetic postmenopausal women (Karimifar et al., 2012; Liu et al., 2023). However, there was a lack of awareness regarding the diagnosis of osteoporosis and the risk of osteoporotic fractures in this cohort, even in developed countries such as the United States (Lewiecki et al., 2019). Postmenopausal women can benefit from additional education about osteoporosis, and our easy-to-use self-assessment tool for osteoporosis may improve awareness among postmenopausal women with T2D and aid in preventing and managing osteoporosis.

Our results indicated that age and BMI were independent predictors of osteoporosis, based on which we successfully developed a tool for categorizing postmenopausal women with T2D into low-, moderate-, and high-risk groups. Our tool is specifically tailored for postmenopausal women. While several tools have been developed to assess osteoporosis or fracture risk, the widely used FRAX algorithm provides 10-year fracture probability for different populations based on 12 risk factors, but does not include diabetes as a factor. Despite considering numerous risk factors, it has been reported that the FRAX algorithm significantly underestimates fracture risk (Roux et al., 2014; Zoccarato et al., 2022). The OSTA (Koh et al., 2001) and OSTC (Tang et al., 2020) tools were designed to identify women at increased risk of osteoporosis by taking into account age and weight. It is important to note that diabetes contributes to an elevated risk of osteoporosis (Schacter & Leslie, 2021). However, both OSTA and OSTC tools did not take diabetes into consideration. Additionally, these tools may assign the same osteoporosis risk to two individuals of the same age and weight but different heights, which may be inappropriate. BMI may be a better predictor of osteoporosis than weight alone, as it more accurately assesses body composition (Asomaning et al., 2006). Our tool incorporates BMI instead of solely relying on weight like OSTA and OSTC. Importantly, our tool was developed using data from postmenopausal women with T2D and has been validated using both the internal and external test datasets.

Age and BMI have been well-documented as significant risk factors of osteoporosis. However, few studies focused on the cohort of postmenopausal women with T2D. Our results indicated that BMI and age were independently associated with BMD at skeletal sites of total hip, femur neck, and lumbar spine L1-L4. They were independent predictors of osteoporosis in postmenopausal women with T2D. Each one-unit increase in BMI resulted in a significant increase in BMD of 12.9, 10.9, and 15.5 mg/cm2 at the total hip, femur neck, and lumbar spine, respectively. The mechanisms of this association include increased mechanical loading and strain, effects of adipokines on bone cells, and higher aromatase activity (Bachmann et al., 2014; Kajimura et al., 2013; Riis, Rødbro & Christiansen, 1986). Although BMI is positively correlated with BMD, it should be noted that the fracture risk at the proximal humerus, distal radius, upper leg, and ankle in obesity is higher, and a large number of low-trauma fractures occur in overweight and obese populations (Asomaning et al., 2006; Premaor et al., 2010; Xu et al., 2017). Therefore, BMI was a crucial modifiable risk factor, and it was advised to maintain a normal BMI for postmenopausal women with T2D. In addition to age and BMI, we found that atherosclerosis was independently associated with lower BMD in the total hip and femur neck. The presence of atherosclerosis can be identified by several methods, including coronary angiography, intravascular ultrasonography, B-mode ultrasonography, CT and MRI (Toth, 2008). Diabetes mellitus can actually induce atherosclerosis development or further accelerate its progression (Poznyak et al., 2020). Osteoporosis and atherosclerosis might share risk factors or pathological mechanisms (Omura, Nishio & Kashiwagi, 2007). A gene analysis (Mo et al., 2022) revealed that immune and inflammatory responses might be a common feature in the pathophysiology of both diseases.

We also discovered a negative correlation between the number of children and BMD, with osteoporosis patients having a higher number of children compared to non-osteoporosis patients. Previous studies have investigated this relationship in postmenopausal women but did not take diabetes into account. A cross-sectional study found that having six or more children was linked to decreased lumbar spine BMD in postmenopausal women (Yang, Wang & Cong, 2022), while another study reported a positive correlation between the number of children and osteoporosis in the univariate analysis but not in the multivariate analysis (Cavkaytar et al., 2015). The transient decrease in BMD associated with lactation was also noted (Lujano-Negrete et al., 2022). Some studies suggested that a history of breastfeeding significantly increased the risk of osteoporosis, while others indicated a potential increase in BMD. In contrast, other studies demonstrated an insignificant relationship between breastfeeding and osteoporosis in postmenopausal women (Yan et al., 2019). However, it is worth noting that the number of children was not independently associated with BMD in our study.

This study has several limitations. First, this was a retrospective study, and we did not perform a longitudinal study of fracture evaluation. Second, our model was developed based on the Chinese population and may not be generalizable to other populations. Further evaluation is necessary to determine its performance in diverse populations. Third, patients only underwent DXA examinations at one time point, which limited our ability to establish temporal associations between risk factors and BMD. However, we have successfully identified the risk factors associated with lower BMD among postmenopausal women with T2D and developed a user-friendly self-screening tool for osteoporosis. It is common for this population to lack awareness of the diagnosis of osteoporosis and their risk of osteoporotic fractures. Our model is specifically tailored for postmenopausal women with T2D and has been validated in both internal and external datasets. Our study may enhance their understanding of osteoporosis and contribute to improved disease management in clinical practice.

Conclusions

In summary, we conducted a quantitative investigation into the correlation between risk factors and BMD. Our findings indicate that age and BMI were independently associated with BMD and were independent predictors of osteoporosis in postmenopausal women with T2D. BMI is a significant modifiable risk factor; thus, we recommend maintaining a normal BMI within this population. Additionally, we have developed an easy-to-use self-assessment tool for early osteoporosis screening based on age and BMI, which may contribute to raising awareness of osteoporosis and prove valuable for disease prevention and management among postmenopausal women with T2D.

Supplemental Information

Supplemental Information 1 Training Dataset for the development of a self-assessment tool for early osteoporosis screening in postmenopausal women with Type 2 diabetes.

Supplemental Information 2 Dataset: test.

Supplemental Information 3 Dataset: validation.

Supplemental Information 4 Model Development code.

Additional Information and Declarations

Competing Interests

Author Contributions

Human Ethics

Data Availability

Yuguo Wei is employed by GE Healthcare. The authors declare that they have no competing interests.

Xiaoyu Chen performed the experiments, prepared figures and/or tables, authored or reviewed drafts of the article, and approved the final draft.

Xiufen Jia performed the experiments, prepared figures and/or tables, authored or reviewed drafts of the article, and approved the final draft.

Junping Lan performed the experiments, authored or reviewed drafts of the article, and approved the final draft.

Wenjun Wu performed the experiments, authored or reviewed drafts of the article, and approved the final draft.

Xianwu Ni performed the experiments, authored or reviewed drafts of the article, and approved the final draft.

Yuguo Wei analyzed the data, authored or reviewed drafts of the article, and approved the final draft.

Xiangwu Zheng conceived and designed the experiments, authored or reviewed drafts of the article, and approved the final draft.

Jinjin Liu conceived and designed the experiments, performed the experiments, analyzed the data, prepared figures and/or tables, authored or reviewed drafts of the article, and approved the final draft.

The following information was supplied relating to ethical approvals (i.e., approving body and any reference numbers):

The ethics committee of the First Affiliated Hospital of Wenzhou Medical University approval to carry out the study (Ethical Application Ref: 2022-r064).

The following information was supplied regarding data availability:

The raw data to develop the self-assessment tool for early osteoporosis screening in postmenopausal women with type 2 diabetes is available in the Supplementary File.

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
