# Peer review of "Association between risk factors and bone mineral density and the development of a self-assessment tool for early osteoporosis screening in postmenopausal women with type 2 diabetes"

_PeerJ, doi:10.7717/peerj.18283_

## Round 0.1 · original submission · Minor Revisions

Dear Dr. Liu and Dr. Zheng,

Your manuscript entitled “Association between risk factors and bone mineral density and development of a self-assessment tool for early osteoporosis screening in postmenopausal women with type 2 diabetes", which you submitted to PeerJ, has been reviewed by the editor and 2 external reviewers.

The reviewers have raised significant concerns that must be addressed before the manuscript can be considered further. However, since the reviewers see merit in your work, I am open to reconsidering the manuscript if you undertake the suggested revisions and resubmit.

If you decide to resubmit the revised version, please summarize all the improvements made in the new version and give answers to all critical points raised in the reviewers’ report in an accompanying letter. Copy and paste each and every reviewer's comment above your response.

Please note that resubmitting your manuscript does not guarantee eventual acceptance. The revised manuscript will undergo a second round of review by the same reviewers. I must emphasize that the acceptability of the revision will depend upon the resolution of the points raised by the reviewers.

Sincerely yours,
Stefano Menini

·

Basic reporting

You have proposed two main objectives for your study. Please define better the second one (line 40)- who is going to use this "tool", how and when . What is the meaning and the aim for "early osteoporosis screening in postmenopausal women with T2D"?

Regarding lines 46 and 52 you are speaking about the same "screening tool"?

Please define clearly how to use the" two straight lines with equations of BMI=0.56*age-4.12 and BMI=0.56*age-10.88".
What is the general message of your paper regarding osteoporosis in T2D population vs general population?

Experimental design

You have collected and analyzed a large amount of data.
In my opinion, the value of article would increase if you will try to present them organized in respect to the proposed main objectives:
- the significance of the analyzed variables in relation with BMD;
- which is the practical value for the two proposed equations for the patients and for general practice in the field of rheumatology, endocrinology and general practice.



There are differences between your data and those coming from general population?
Please define "atherosclerosis" in your study. The T2D is often associated with subclinical atherosclerosis.

Validity of the findings

There is rigorous approach to the collected data.

Please explain clearly how the "two straight lines with the same slope k and
different values of c can divide patients into three risk regions with low-, moderate- and
high prevalence of osteoporosis". lines 158-160

Please detail how do you intend to use the proposed tool in T2D population .There are differences vs general population?

Additional comments

It is very hard to follow the ideas in this articles. The literature review is ambiguous in respect to the objectives of the study (type 2 diabetes population and osteoporosis).I think that minor changes in the manner of define and present the data would increase the value of the article, which is evidently based on a hard work of the team mentioned.

·

Basic reporting

See below

Experimental design

See below

Validity of the findings

See below

Additional comments

Thank you for asking my opinion ‎about ‎the ‎manuscript ‎entitled ‎‎"Association between risk factors and bone mineral density and development of a self-assessment tool for early osteoporosis screening in postmenopausal women with type 2 diabetes". ‎

‎I believe that this manuscript should ‎be ‎major revision:‎

Q1. It is very important to change and modify the title. the ‎title ‎is not appropriate.‎
Q2. Are the objectives and the rationale of the study ‎clearly ‎stated? ‎
‎Q3.‎ In the abstract, the research gap was not clearly ‎stated. In ‎addition, the authors need to rewrite the study ‎objectives ‎to be more academic writing
‎Q4. In the introduction, include the ‎study's significance ‎and ‎novelty. What makes the study different ‎from ‎the rest ‎and ‎what ‎does it add to the current knowledge?‎.‎
Q5. In the introduction, the authors ‎should ‎have ‎explained ‎the ‎purpose of this study and the ‎existing gaps ‎in ‎this field ‎and ‎explained why this study was ‎conducted.‎
‎Q6. Are the methods clear and replicable? Do all the ‎results presented to match the methods described?‎
Q7. If relevant are the results novel? Does the study ‎provide an advance in the field? Is the data plausible?‎
Q8. References are relevant, correct, and not ‎recent. ‎The ‎number ‎of ‎references should be increased.‎ ‎please add some ‎references. since this is a scientific ‎review, all the sentences need ‎to be supported with ‎references.‎
This study is very beautiful. I liked the sequence and ‎enjoyed ‎reading. Please add more references on similar ‎studies.‎
‎Q9. There are a lot of grammatical errors. This ‎must ‎be ‎taken ‎care ‎of and addressed.‎
‎Q10. What are the limitations of the study?‎ A description ‎of ‎limitations is missing at the end of the discussion ‎section.

---

## Round 0.2 · accepted · Accept

Dear Dr. Liu and Dr. Zheng,

Thank you for submitting the revised version of your manuscript. After a thorough review of the changes by the reviewers and myself, I am pleased to inform you that all the reviewers' comments have been adequately addressed. Therefore, your manuscript is ready for publication in its current version in PeerJ.

I thank all reviewers for their efforts in improving the manuscript and the authors' cooperation throughout the review process.

Sincerely yours,
Stefano Menini

·

Basic reporting

No comment

Experimental design

No comment

Validity of the findings

No comment

Additional comments

I have read the new version of the article with the changes made by the authors, which address all the questions and suggestions from my initial review. Therefore, I propose the publishing of the article in the revised form.